# Predicting the 14-Day Hospital Readmission of Patients with Pneumonia Using Artificial Neural Networks (ANN)

**DOI:** 10.3390/ijerph18105110

**Published:** 2021-05-12

**Authors:** Shu-Farn Tey, Chung-Feng Liu, Tsair-Wei Chien, Chin-Wei Hsu, Kun-Chen Chan, Chia-Jung Chen, Tain-Junn Cheng, Wen-Shiann Wu

**Affiliations:** 1Pulmonary Medicine, Chi-Mei Medical Center, Tainan 700, Taiwan; bbfarn02@gmail.com; 2Department of Medical Research, Chi-Mei Medical Center, Tainan 700, Taiwan; chungfengliu@gmail.com; 3Department of Pharmacy, Chi-Mei Medical Center, Tainan 700, Taiwan; u103534001@gmail.com; 4Division of Clinical Pathology, Chi-Mei Medical Center, Tainan 700, Taiwan; 990129@mail.chimei.org.tw; 5Department of Information Systems, Chi-Mei Medical Center, Tainan 700, Taiwan; carolchen@mail.chimei.org.tw; 6Departments of Neurology and Occupational Medicine, Chi-Mei Medical Center, Tainan 700, Taiwan; tjcheng@mail.chimei.org.tw; 7Division of Cardiovascular Medicine, Chi-Mei Medical Center, Tainan 700, Taiwan; 8Department of Pharmacy, Chia-Nan University of Pharmacy and Science, Tainan 700, Taiwan

**Keywords:** unplanned patient readmission, artificial neural network, convolutional neural network, nurse, Microsoft Excel, receiver operating characteristic curve

## Abstract

Unplanned patient readmission (UPRA) is frequent and costly in healthcare settings. No indicators during hospitalization have been suggested to clinicians as useful for identifying patients at high risk of UPRA. This study aimed to create a prediction model for the early detection of 14-day UPRA of patients with pneumonia. We downloaded the data of patients with pneumonia as the primary disease (e.g., ICD-10:J12*-J18*) at three hospitals in Taiwan from 2016 to 2018. A total of 21,892 cases (1208 (6%) for UPRA) were collected. Two models, namely, artificial neural network (ANN) and convolutional neural network (CNN), were compared using the training (*n* = 15,324; ≅70%) and test (*n* = 6568; ≅30%) sets to verify the model accuracy. An app was developed for the prediction and classification of UPRA. We observed that (i) the 17 feature variables extracted in this study yielded a high area under the receiver operating characteristic curve of 0.75 using the ANN model and that (ii) the ANN exhibited better AUC (0.73) than the CNN (0.50), and (iii) a ready and available app for predicting UHA was developed. The app could help clinicians predict UPRA of patients with pneumonia at an early stage and enable them to formulate preparedness plans near or after patient discharge from hospitalization.

## 1. Introduction

Unplanned patient readmission (UPRA) continues to attract considerable attention because of its substantial negative influence on patients’ quality of life and healthcare costs [1]. More than 8818 articles searched using the keyword “patient readmission” (MeSH Major Topic) were found in the PubMed library [2]. From July 2015 to June 2016, 15.2% of Medicare beneficiaries experienced UPRA within 30 days after discharge [3]. UPRA has been estimated to account for $17.4 billion in Medicare expenditure annually [4], and a total of 3.3 million (more than 55%) of patients are on Medicare [5].

### 1.1. Related Work

#### 1.1.1. Hospital Readmissions Are Harmful to Patients

Hospital readmissions are harmful to patients [6]. Older adult readmissions are mostly associated with delirium, frailty, and a significant decline in functional ability, resulting in disability and loss of independence [7]. Of the more than 7 million readmissions annually, 836,000 are estimated to be avoidable [8]. Thus, healthcare quality needs to be improved and UPRA rates need to be decreased.

The Affordable Care Act [9] implemented the Hospital Readmission Reduction Program (HRRP) [10] in 2012 to use 30-day UPRA as a metric to financially penalize hospitals with excessive UPRA rates. The high associated cost and penalty strategy of the HRRP have intensified the efforts of different healthcare settings in reducing their UPRA rates.

#### 1.1.2. Traditional Solutions Required for Hospital Readmissions

Traditional solutions to mitigate UPRA merely focus on passively complementing in-patient care with enhanced care transition and post-discharge interventions. Nonetheless, evidence has shown that UPRA is related to inadequate or substandard in-patient care, such as premature discharge [11] and inferior nosocomial (hospital-acquired) infection [12]. Thus, interventions are resource-intensive [13], and no single intervention or bundle of interventions can be significantly effective [14]. Regrettably, the traditional interventions hardly improve the quality of in-patient care because they are initiated near or after discharge when clinicians’ role in in-patient care is close to ending.

#### 1.1.3. Modern Prediction Models Used for Hospital Readmissions

Alternatively, modern predictive modeling is an efficient method to reduce UPRA because it stratifies patients’ readmission risk and targets preventive interventions to patients at high risk [15]. Numerous models for the early detection of UPRA have been reported; however, their performance (i.e., accuracy and stability) and design (i.e., usefulness and feasibility) are unsatisfactory. For instance, Wang et al. [16] developed a real-time model using the time series of vital signs and discrete features, such as laboratory tests. However, this model’s prediction accuracy was not sufficiently high (area under the receiver operating characteristic curve (AUC) = 0.70) [17] to deploy the model in the hospital information system with the proposed forecasting algorithms to support treatment because many false-positive cases appear in these imbalanced-class data [18,19,20,21], increasing the clinicians’ burden.

Benjamin et al. [22] developed a laboratory-based model specific to patients with heart failure within 24 h of admission; however, the performance of the model was poor, with AUCs of 0.57 and 0.59 in the female and male validation sets, respectively. Patrick et al. [23] reported an early detection model based on the information available at admission and in the index admission medication record with a moderate performance (AUC = 0.67) in the validation sets. Furthermore, none of the previously mentioned studies excluded planned readmissions following the CMS guideline [24]. The two issues of model performance (i.e., accuracy and stability) and design (i.e., usefulness and feasibility) should be simultaneously taken into account. 

#### 1.1.4. Topic Selection in Pneumonia

Pneumonia is the most common reason for UPRA within 30 days after discharge [6]. The measures for evaluating UPRA are risk-standardized mortality, risk-standardized readmission, and excess days in acute care [3].

The incidence of pneumonia in 2011 was approximately 157,500 [25]. The annual incidence of new cases is 150.7 million, of which 11–20 million (7% to 13%) are severe enough to require hospital admission [26]. Approximately 90% of pneumonia cases occur while patients are mechanically ventilated in intensive care units [27]. Pneumonia increases hospital length of stay by 7–9 days, has a crude mortality rate of 30% to 70%, and is associated with an estimated cost of $40,000 or more per patient [28]. Moreover, pneumonia has been shown to develop in 9% to 40% of patients after abdominal surgery, with an associated mortality rate of 30% to 40% [29]. Thus, we are motivated to build a prediction model for the early detection of UPRA of patients with pneumonia.

### 1.2. Study Objectives

This study aimed (i) to build a prediction model for the early detection of UPRA of patients with pneumonia using the machine learning technique and (ii) to develop a system, such as an app, that can continuously monitor readmission risk during hospitalization.

## 2. Materials and Methods

### 2.1. Study Sample and Demographic Data

This study is a retrospective analysis of electronic health record (EHR) data. In contrast to most models proposed in previous studies that focus only on index admission characteristics, we included a detailed medical history of previous encounters up to 1 year before index admissions to construct a better prediction model for UPRA.

We downloaded 25,385 records of in-patient data, including those with pneumonia as a primary disease (i.e., those with ICD-9:480*-486 and ICD-10:J12*-J18*). Based on the CMS guideline [24], planned readmissions were excluded. A total of 21,892 eligible cases (1208 (6%) for UPRA) without missing data were collected. Adult patients (age ≥ 20 years) with pneumonia in three hospitals (i.e., Chi Mei Medical Center, Chi Mei Liouying Hospital, and Chi Mei Chiali Hospital with 1200, 600, and 200 beds, respectively, and 11,026, 6260, and 4606 cases, respectively) between 2016 and 2018 were identified and included in this study.

To ensure that our prediction model for UPRA works early during hospitalization, we only used index admission attributes for which the values are available in EHR data, including patients’ demographics, laboratory tests, vital signs, and medications. Patients’ data were enriched by a detailed history of previous hospital encounters within 1 year before the current in-patient stay, including information about the diagnosis, procedure, laboratory tests, vital signs, medications, and healthcare utilization.

This study was approved and monitored by the institutional review board of Chi Mei Medical Center (Taiwan; 10910-00). All hospital and participant identifiers were stripped.

### 2.2. Feature Variables (Task 1)

We established a multidisciplinary team, including physicians and specialists in pneumonia, data scientists, information engineers, nurse practitioners, and quality managers, for this study on artificial intelligence implementation. The criteria for the inclusion of cases were determined by the multidisciplinary team. Patients who did not have a record of contracting pneumonia were excluded.

Feature variables were extracted from 57 items that were determined by the multidisciplinary team using the Weka software [30] via the following steps: (i) standardize each variable to the mean (0) and standard deviation (i.e., SD = 1), (ii) use the search method (Select Attributes)/(InfoGainAttributeEval) (Attribute Evaluator)/(Ranker)(Search Method), (iii) use full training sets, and (iv) click on the suggested feature items.

Forest plots [31,32,33] were drawn to present the extracted feature variables. One plot compares the ratios in counts of events and nonevents within two groups (i.e., UPRA and non-UPRA) using the odds ratio method, similar to the traditional Chi-square test. Another plot is similar to the traditional *t* test for continuous variables. Notably, all continuous-type data were transformed into standardized scores ((observed scores − mean)/standard deviation [SD] × 1.7), where 1.7 is the adjustment factor from normal standard distribution to logistic distribution [34,35,36,37]. The standard mean difference (SMD) method was utilized to compare the differences in variables alone (such as the t test) and with hospital types (such as an analysis of variance) using the forest plot.

The Chi-square test was conducted to assess the heterogeneity between variables. The forest plots (confidence interval (CI) plot) were drawn to display the effect estimates and their CIs for each study.

### 2.3. Model Building and Scenarios in Comparison (Task 2)

We focused on model accuracy (e.g., >0.7) and stability (or generalizability, e.g., the discrepancy between training and test sets) out of various perspectives, such as model feasibility, efficacy, and efficiency, using the maximum AUC between models using the training cases to predict the learning cases; see the following steps to create the prediction models and design the scenarios in comparison:

#### 2.3.1. Models in Comparison

The artificial neural network (ANN) and convolutional neural network (CNN) were analyzed using the four scenarios previously mentioned. The CNN has traditionally been performed on Microsoft (MS) Excel (Microsoft Corp., New York, NY, USA) [38,39,40,41]. As illustrated in Figure 1, the ANN process involves data input in Layer 1, where the data are combined with two types of parameters and run through the sigmoid function algorithms in Layers 2 and 3. Finally, as shown on the right side and bottom of Figure 1, the prediction model was deemed complete when the total residuals were minimized using the MS Excel function of SUMXMY2 and Solver add-in. 

#### 2.3.2. Scenarios in Comparison

First, the 21,892 participants were randomly split into training and test sets in a proportion of 70% (*n* = 15,324) to 30% (*n* = 6568), where the training set was used to predict the test set. 

Second, the accuracy (e.g., SENS, SPEC, and AUC) and stability (or generalizability, e.g., using the training set to predict the test set evaluated by observing the AUC as well) were verified (e.g., AUC > 0.70). The training and test sets are provided in Appendix A.

### 2.4. Data Presentations in Results

#### 2.4.1. Presentation 1: Comparison of Accuracy on Two Models

Accuracy was determined by observing the high AUC along with indicators of SENS, SPEC, and accuracy in both models. The definitions are listed as follows:True positive (TP) = the number of predicted UPRA to the true UPRA(1)
True negative (TN) = the number of predicted Non-UPRA to the true Non-UPRA(2)
False positive (FP) = the number of Non-UPRA − TN(3)
False negative (FN) = the number of UPRA − TP(4)
Sensitivity (SENS) = true positive rate (TPR) = TP ÷ (TP + FN)(5)
Specificity (SPEC) = true negative rate (TNR) = TN ÷ (TN + FP)(6)
ACC = accuracy = (TP + TN) ÷ N(7)
N = TP + TN + FP + FN(8)
AUC = (1 − Specificity) × Sensitivity ÷ 2 + (Sensitivity + 1) × Specificity ÷ 2(9)
SE for AUC = √ (AUC × (1 − AUC) ÷ N)(10)
95% CI = AUC ± 1.96 × SE for AUC(11)
Accuracy rate = (TP + TN) / (TP + TN + FP + FN)(12)

#### 2.4.2. Presentation 2: Comparison of Prediction Models Referring to Algorithms in Weka Software

To better understand the effectiveness and efficacy of the ANN and CNN models, several machine learning algorithms in the Weka software (University of Waikato, Wellington, New Zealand) were illustrated to compare the high indicators of SENS, SPEC, accuracy, and AUC between ANN and CNN models.

All indicators are based on high AUC rather than the accuracy in Equation (12). It is because imbalanced-class data exist in this study (e.g., 1208 (6%) for UPRA vs. 20,684 (94%) for non-UPRA). High accuracies rates with imbalanced SENS and SPEC are expected in imbalanced-class data using the traditional approaches [18,19,20,21]. Thus, we applied the minimization of average model residuals in both classes (i) to obtain balanced SENS and SPEC and (ii) to overcome the disadvantage of high accuracy rates (i.e., the minimum residuals minimized by the formula of average (residuals in UPRA) + average(residuals in non-UPRA)). It is hard to gain balanced SENS and SPEC using professional machine-learning software when an imbalanced number of classes exists, unless the method of minimizing model residuals is controlled by the user. 

#### 2.4.3. Presentation 3: Developing an App for Predicting UPRA (Task 3)

An app for the early detection of all-cause 14-day UPRA of patients with pneumonia was designed and developed because the penalty strategy of the Taiwanese government-run health insurance administration (TGHIA) forced many hospitals in Taiwan to reduce the 14-day UPRA. Model parameters were embedded in the computer module. The results of the classification (i.e., UPRA and non-UPRA) instantly appear on smartphones. The visual representation with binary (i.e., UPRA and non-UPRA) categorical probabilities is shown on a dashboard displayed on Google Maps.

#### 2.4.4. Caution in Estimation of Model Parameters (Task 4)

Points of caution were addressed to improve the model accuracy and AUC under the scenario of imbalanced-class data. For instance, an example consists of 1000 cases (*n* = 100 and 900 for UPRA and non-UPRA, respectively). A highly accurate rate reaches 0.90 assuming that all cases are classified as Non-UPRA. However, the SENS and SPEC are 0.0 and 1.0, respectively. The AUC equals 0.5 ($\left(0\times \left(1-1.0\right)÷2+\left(0+1.0\right)\times 1.0÷2≅0.5\right)$ based on Equation (9). As such, the AUC is considered in this study to compare the model accuracy and stability between prediction models. 

### 2.5. Statistical Tools and Data Analysis

IBM SPSS Statistics 22.0 for Windows (SPSS Inc., Chicago, US) and MedCalc 9.5.0.0 for Windows (MedCalc Software, Ostend, Belgium) were used to obtain the descriptive statistics and frequency distributions among groups and to compute the model prediction indicators expressed in Equations (1)–(12). The significance level of type I errors was set at 0.05. ANN and CNN were performed on MS Excel.

A visual representation of the classification was plotted using two curves based on the probability theory of the Rasch model [42]. Four tasks of data representations are involved in obtaining the results; see the study flowchart in Figure 2. The ANN modeling process with an MP4 video is provided in Appendix B [43].

## 3. Results

### 3.1. Task 1: Feature Variables Extracted from the Data

Of the original 57 items, 17 feature variables were extracted using the Weka software. Figure 3 [44] and Figure 4 [45] show the odds ratios and SMD methods used in the meta-analysis, respectively [31,32,33]. The series of numbers before the variables are the order assigned by the Weka software, with the most significant importance for the binary classification in machine learning.

Figure 3 shows that 13 variables (of them, 11 closer to the right side) are statistically different in frequency between the UPRA and non-UPRA groups. The two other variables (i.e., hospitals A and B) that are closer to the left side have a lower frequency of UPRA than hospital C that favors the right side, indicating more UPRA at hospital C. The Q-statistic is 413.63, with degrees of freedom = 12 (*p* < 0.001), indicating that the odds ratios of the 13 variables are significantly different.

Similarly, all variables but one (i.e., abnormal CRP frequency during hospitalization, with the correlation coefficient = 0.007) have a significant tendency to favor the UPRA side, as shown in Figure 4. The variable “doctor age” favors the left side, indicating that younger physicians have a higher number of UPRA cases within 14 days after discharge from hospitalization than older physicians, with a negative correlation coefficient (−0.04).

The Q-statistic is 409.41, with degrees of freedom = 5 (*p* < 0.001), indicating that the SMDs between the UPRA and non-UPRA groups corresponding to the six variables are significantly different.

If hospital types are considered in the comparison of differences between doctor age, hospital C (14) favors the left side, as shown in Figure 5. No difference in variables was observed among hospital types in Figure 5. Readers are invited to scan QR-codes in Figure 3, Figure 4 and Figure 5 for detailed information on internet.

### 3.2. Task 2: Comparisons of Accuracies in Training and Test Samples

When comparing the two models with the data set of 15,324 cases, the ANN model has a higher AUC than the CNN, indicating that the ANN model has higher (i) accuracy (i.e., 0.75:0.51) and (ii) stability (0.73:0.50) than the CNN model (see the AUC in Table 1). 

Notably, the accuracies in WeKA fail to construct a balanced SENS and SPEC. The high accuracy is problematic and unreliable due to a tendency to favor non-UPRA classification for all cases (i.e., none was classified as UPRA due to the imbalanced-class numbers in the data. As such, the high accuracy in Weka is meaningless. We should consider the composite score of AUC in the evaluation of model validation. Furthermore, the stability (i.e., using the training cases to predict the test cases) cannot be obtained through the Weka tool. Readers are invited to verify the results in Weka by administrating the data of the training and test sets provided in Appendix A (refer to the MP4 video in Appendix B).

### 3.3. Task 3: Web-Based Assessment of the App for Predicting UPRA

The interface of the app for predicting UPRA within 14 days after discharge for patients with pneumonia is shown on the left-hand side of Figure 6. Readers are invited to click on the links [46,47] and to interact with the UPRA app; see Appendix C. Notably, all 53 model parameters are embedded in the 17-item ANN model. Once the responses [45] are submitted, the app generates a result (shown on the right-hand side of Figure 6) as a classification of either possible UPRA or non-UPRA on smartphones.

An example in which the patient scored a high probability (0.94) of UPRA is shown on the right-hand side of Figure 6. The curve starts from the bottom-left corner to the top-right corner. The sum of the probabilities of UPRA and non-UPRA is 1.0. The odds ratio can be calculated using the formula *p*/(1 − *p*) (0.95/0.05 = 15.67), indicating that this discharged patient has a high probability of UPRA within the next 14 days.

### 3.4. Task 4: Cautions Addressed in Estimation of Model Parameters

Due to the imbalanced-class data in the current study, we created Figure 7 to illustrate the use of our readmission prediction model that works well early during patient hospitalization. However, imbalanced SENS and SPEC are observed. A high TPR leads to a high FNR, as shown in Figure 7C. Due to imbalanced-class numbers in the two groups, a high accuracy (e.g., 0.93 = (15,324 − 1000)/15,324 in Table 1, where 1000 is the number of UPRA in the training set) is obtained for all cases classified as non-UPRA, as shown in Figure 7A. Otherwise, a medium accuracy would be in Figure 7B.

To overcome this problem, a scheme called matching personal response scheme to adapt for correct classification in the model (MPRSA) [38] was used to reduce the number of false-positive cases in the non-UPRA group and to ensure that the model’s accuracy reaches 100%. The reason for using MPRSA is that the known patterns and their corresponding labels (i.e., UPRA or non-UPRA) used as a reference in the model for predicting unknown labels have a high accuracy and prevent the ANN from failing in the classification of the known responses. 

Detailed information about the MPRSA scheme that can reduce the burden of false-positive cases at the early stage in detecting 14-day UPRA of patients with pneumonia is based on Reference [39]. Thus, all false alerts are reduced to as few as possible.

## 4. Discussion

### 4.1. Principal Findings

We observed that (i) the 17 feature variables extracted from 57 items in this study using the ANN model yielded a higher AUC (0.75) than the CNN models and that (2) the ANN exhibited better prediction accuracy (0.73 in stability denoted by AUC) than the CNN, and (3) a ready and available app for predicting UPRA with a link that can be provided to readers was developed.

### 4.2. What This Finding Adds to What We Already Knew

#### 4.2.1. Literature Reviews of Feature Variables

The most frequent primary diagnoses in early readmissions were pneumonia (shown as #14 in Figure 3), heart failure, chronic obstructive pulmonary disease (COPD), and sepsis [3,6,48,49]. 

The 17 UPRA predictors related to medical history and index admission were extracted in this study. The results are similar to those in [50,51,52] for (1) male sex (shown as #11 in Figure 4), three or more previous admissions (shown as #1 in Figure 4), chronic lung disease, and cancer (shown as #16 in Figure 3); (2) length of stay in days (shown as #2 in Figure 4), COPD (shown as #5 in Figure 3), and age (shown as #5 in Figure 4); and (3) platelets (shown as #13 in Figure 4), utilization history ≥ 1, hospitalizations in the past year (shown as #1 in Figure 4), age (shown as #5 in Figure 4), and male sex (shown as #11 in Figure 4).

Patients treated with chemotherapy in the previous year were more associated with readmission than patients not treated with chemotherapy. This finding can be explained by the link between chemotherapy (shown as #16 in Figure 3) and cancer, which has been reported as a predictor of readmission [53,54,55].

Blood disorder or an abnormal amount of a substance in the blood (shown as #12 and #13 in Figure 4) can indicate certain diseases or side effects. Having an increased number of abnormal test results indicates that the patient is frail and is prone to readmission.

The prescription of two medications (shown as #10 and #15 in Figure 3) was observed to be positively linked to UPRA. These medications may have side effects that are associated with UPRA. COPD (shown as #5 in Figure 3) has been reported as a risk factor of readmission [55]. Interestingly, the prescription of antibiotic_rear (shown as #15 in Figure 3) in previous encounters and index admission is positively associated with UPRA. One possible explanation is that antibiotic_rear used to treat infections caused by bacteria can potentially cause UPRA within 14 days after discharge [56,57].

#### 4.2.2. Comparison of Variables in Different Count Events in Two Groups

The 17 UPRA predictors can be classified into 2 categories, namely, binary and continuous variables, using the forest plots [31,32,33] to display a difference that is similar to that for traditional methods using the Chi-square and *t*-tests to identify the discrepancy in the number of events and means between two groups [48,49]. The results shown in the forest plots are equivalent to the method using multiple logistic regression.

#### 4.2.3. Comparison of Model Accuracies in the Literature

A systematic review of model performance for predicting the risk of UPRA for patients with pneumonia [53] shows that model discrimination (C-statistic or AUC) ranged from 0.59 to 0.77 (median = 0.63), similar to our study results shown in Table 1.

### 4.3. Contributions from This Study

#### 4.3.1. ANN Module Developed on MS Excel

ANN [58,59] was performed on MS Excel, which has not been reported in the literature. An app was designed to display the classification results using the categorical probability theory in the Rasch model [42]. The animation-type dashboard was incorporated into the ANN model to enable easy understanding of the classification results with visual representations.

#### 4.3.2. The Imbalanced-Class Data Considered in Estimation of Model Parameters

The different types of algorithms for classification in machine learning [60,61] are logistic regression, support vector machine [61], naïve Bayes, random forest classification, ANN, CNN [38,39,40,41], and k-nearest neighbor [61]. ANN was superior to the other algorithms, with a 93.2% classification accuracy in a previous study [60]. However, accuracy of the application of ANN in the prediction of UPRA is not high (e.g., AUC between 0.55 and 0.65) according to a previous study [62].

In general, large population-based or multicenter models exhibit poor performance. The nine studies included in a review of risk prediction models for hospital readmission [63] had AUCs between 0.55 and 0.65. However, our UPRA ANN prediction model has better discriminability (AUC = 0.73 in stability) than other machine learning algorithms shown in Table 1. The caution mentioned in this study is the imbalanced-class data considered in the estimation process of model parameters.

We applied the minimization of average model residuals in both classes to obtain balanced SENS and SPEC and to overcome the disadvantage of high accuracy rates. Imbalanced-class data resulting in a high accuracy are demonstrated in Table 1 and were overcome in this study when considering balanced residuals in model optimization. 

#### 4.3.3. An App Developed to Predict the UPRA Using Online Visualization 

We built an app to display the results using the visual dashboard on Google Maps. The animation-type dashboard was incorporated in the ANN model to enable readers to understand the classification results with visual representations and to practice it on their own with links [43,44,45,46,47], [64,65], which has not been reported in the literature (e.g., only comparisons between model accuracies were presented in some studies [23,24,25,66]). As a result, the app evidently (Figure 5 [42,43,44,45,46]) enables point-of-care prediction that can be used to continuously monitor UPRA risk during the entire duration of hospitalization.

#### 4.3.4. The Forest Plot Used to Interpret the Feature Variables 

Traditionally, feature variables are listed in a table rather than as visualizations, such as we present within the app using a forest plot for ease of comparison between variables [42,43,64,65], which has never been used before within the topic of machine learning. 

### 4.4. Implications and Future Work

The ANN exhibited better accuracy and stability than the CNN in this study. To our knowledge, no other study has used the ANN approach to predict UPRA, which is a breakthrough in this study and no studies have incorporated indicators of accuracy and stability to verify model feasibility, efficacy, and efficiency, although several authors have used the split scheme with a 70:30 ratio to validate their predictive CNN models [37,38,39].

More than 2062 articles searched using the keyword “artificial neural network” (title) were found in PubMed Central on 10 October 2020. None of the articles used MS Excel to perform the ANN. The interpretations of the ANN concept and process as well as the parameter estimations, are shown in Figure 1, Appendix B, and the app [41,42,43,44,45]. Readers can estimate the parameters in the ANN model on their own and can examine the differences between their results and that from the current study.

In addition to the performance of the ANN model (i.e., AUC = 0.73), we considered its generalizability. To ensure good generalizability, the MPRSA scheme [38] was utilized to avoid imbalanced numbers in the UPRA (6%) and non-UPRA (94%) groups in this study and to ensure that the model’s accuracy reaches 100% without increasing clinicians’ burden resulting from false-positive cases during the prediction of UPRA.

The categorical probability curves are shown in Figure 6. The binary categories (e.g., success and failure of an assessment in the psychometric field) have been frequently applied in health-related outcomes [37,38,39], [66,67]. However, we are the first to provide categorical probability curves of the UPRA animation-type dashboard displayed on Google Maps (Figure 6).

### 4.5. Limitations and Suggestions

Although our model was designed to be specific to patients with pneumonia, it does not work for patients under 20 years old and patients outside the investigated hospitals (i.e., A, B, and C). The reason for this is that infant and pediatric readmissions were reported to have different patterns from adult readmissions [58,68] and could be influenced by parental factors [69,70].

Next, although the 17-item UPRA has been validated, there is no evidence to support that the item “whether abnormal CRP frequency exists during this hospitalization” shown in Figure 4 (Z = 1.12, *p* = 0.263, similar to multiple logistic regression) between the UPRA and non-UPRA groups should be removed. Thus, this item (#12 in Figure 4) selected by the Weka software should be verified further in the future.

Third, we did not discuss possible further improvements in predictive accuracy. For instance, whether other feature variables (e.g., variables not shown in Figure 3 and Figure 4) should be applied to the ANN model to increase the accuracy rate is worth discussing. In the future, it would be useful to look for other variables that can improve the power of the UPRA prediction model.

Fourth, the study was performed using the ANN model. Whether other prediction models not illustrated in Table 1 have higher accuracy and stability than the ANN model has yet to be investigated.

Fifth, many articles investigated the factors for 30-day readmission to hospitals [71]. Few articles applied 14-day UPRA to build a prediction model. The reason we use the 14-day UPRA is due to the penalty strategy launched by the TGHIA in all hospitals in Taiwan. The results of this study can be generalized to other disparate days (e.g., 30 days) of UPRA.

Finally, the study patients were taken from three types of hospitals (i.e., A, B, and C representing a medical center, a regional hospital, and a local hospital, respectively) in Taiwan. The model parameters estimated for the 14-day UPRA are only suitable for Chinese (particularly Taiwanese) healthcare settings because geolocation is associated with socioeconomic status, which has been reported to be linked to UPRA [72].

Thus, generalization of these UPRA findings (e.g., the model parameters) should be made with caution because the sample only included patients with pneumonia aged ≥20 years in Taiwan. Additional studies in other countries are required in the future to reexamine the feature variables that are similar to those used this study.

## 5. Conclusions

In this study, the ANN was performed on MS Excel. The MPRSA was recommended to increase the model’s prediction accuracy. A ready online app was built to display the results using the visual dashboard on Google Maps. The categorical probability curves based on the Rasch model are unique compared to previous machine-learning studies. Our novel app with our ANN algorithm improves the accuracy of predicting UPRA up to AUC = 0.73. The integration of this app into a hospital information system would be beneficial in minimizing penalization of excessive UPRA rates in the discernible future.

## Figures and Tables

**Figure 1 ijerph-18-05110-f001:**
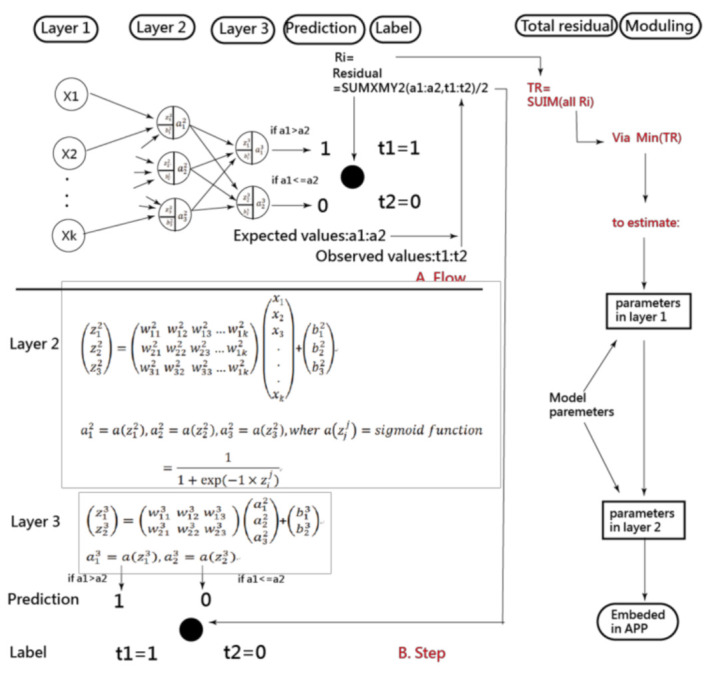
Process of estimating parameters in the ANN model.

**Figure 2 ijerph-18-05110-f002:**
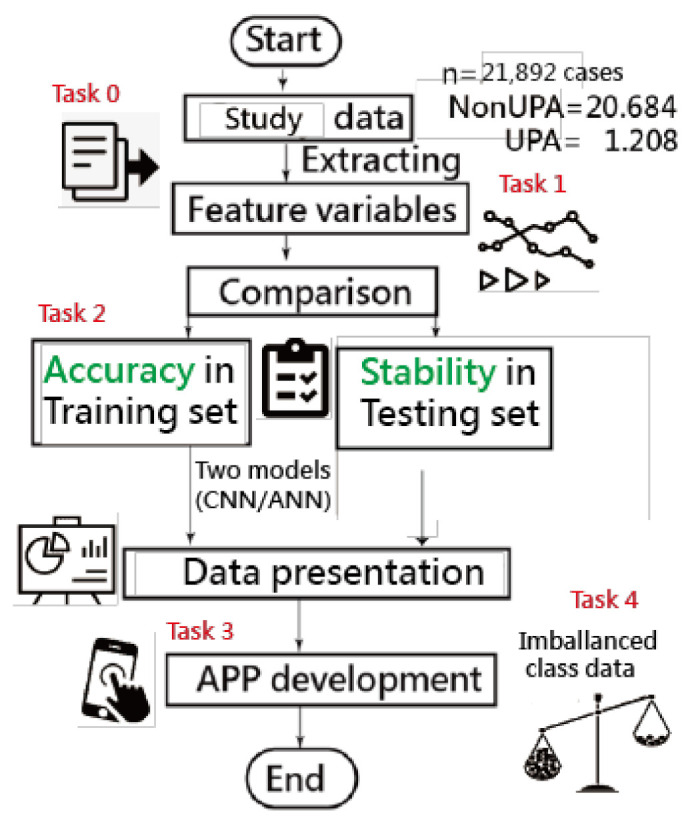
Study flowchart (four major tasks to achieve).

**Figure 3 ijerph-18-05110-f003:**
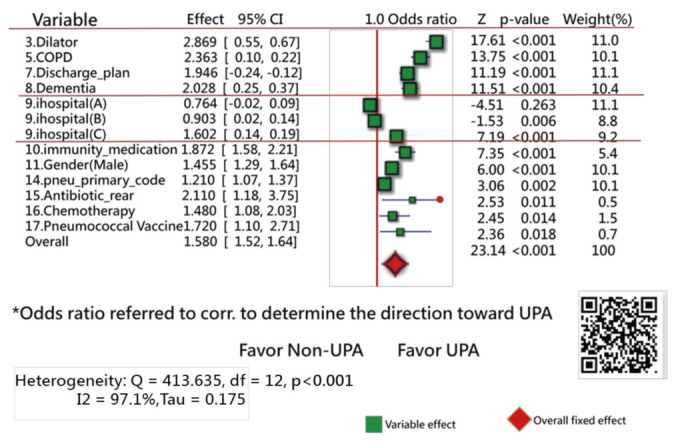
Feature variables using a forest plot to present the interpretation based on the odds ratio method (1).

**Figure 4 ijerph-18-05110-f004:**
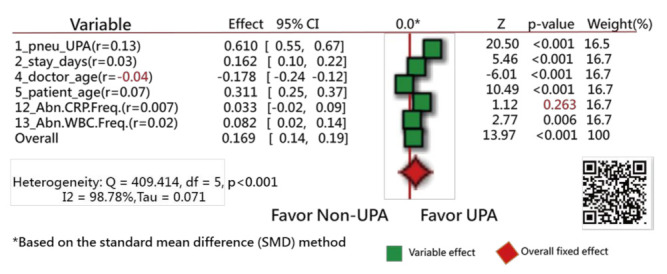
Feature variables using a forest plot to present the interpretation based on the standard mean difference (SMD) method (2).

**Figure 5 ijerph-18-05110-f005:**
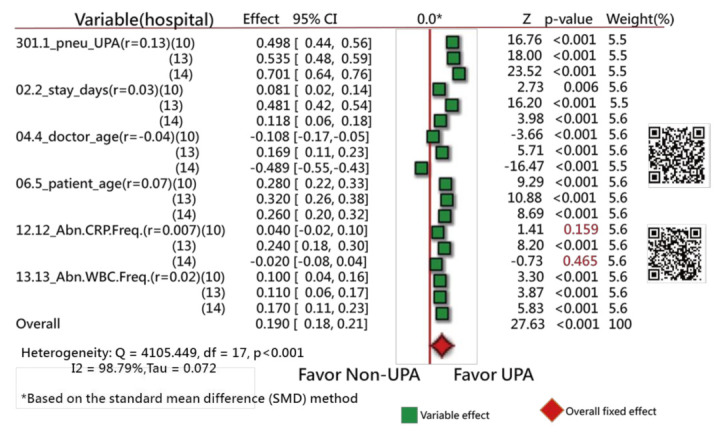
Comparison of hospital types between feature variables using a forest plot to present the interpretation based on the SMD method (3).

**Figure 6 ijerph-18-05110-f006:**
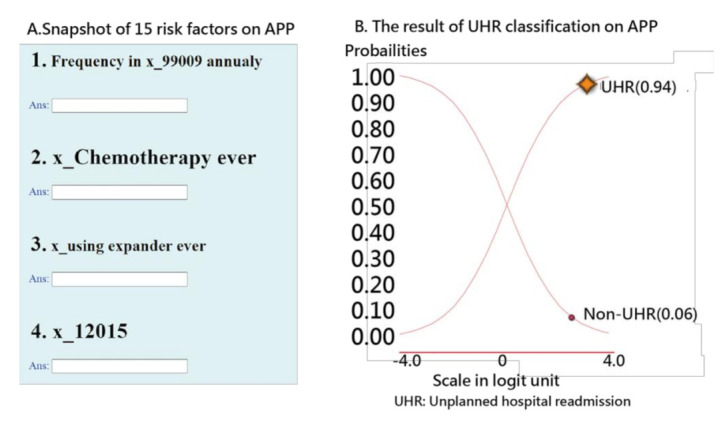
Snapshot of the UPRA app on a smartphone.

**Figure 7 ijerph-18-05110-f007:**
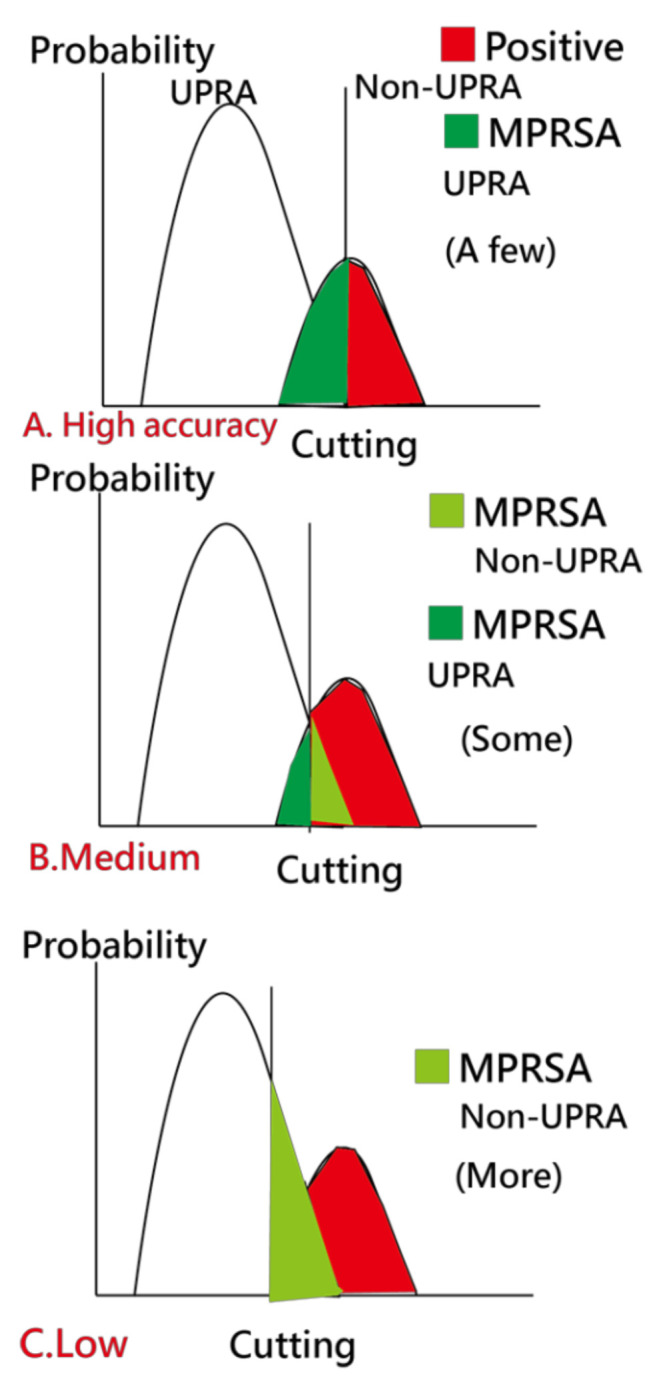
Analysis of the MPRSA strategy.

**Table 1 ijerph-18-05110-t001:** Comparison of statistics in models for accuracy and stability using AUC in the evaluations.

		Training Set				Testing Set			
Model	*n1*	SENS	SPEC	ACC	AUC	SENS	SPEC	ACC	AUC
A: Machine learning algorithms in the Weka software (based on maximum accuracy)									
BayesNet	15,324	0.00	1.00	0.93	0.50				
Logistic	15,324	0.00	1.00	0.93	0.53				
NaiveBayes	15,324	0.01	0.99	0.93	0.53				
SMO	15,324	0.00	1.00	0.93	0.50				
RandomForest	15,324	0.00	1.00	0.93	0.50				
MultiLayer	15,324	0.00	1.00	0.93	0.63				
REPTree	15,324	0.00	1.00	0.93	0.50				
JRIP	15,324	0.00	1.00	0.93	0.50				
LinSVM	15,324	0.00	1.00	0.93	0.50				
J48 (Tree)	15,324	0.00	1.00	0.93	0.50				
B. CNN & ANN	*n1 n2*								
CNN	15,324/6568	0.80	0.21	0.24	0.51	0.88	0.10	0.13	0.50
ANN	15,324/6568	0.80	0.70	0.70	0.75 *	0.69	0.77	0.77	0.73

* AUC = $\left(0.80\times \left(1-0.70\right)\right)÷2+\left(0.80+1.00\right)\times 0.70÷2=0.75;$
n1:training sample size; n2:testing sample size.

## Data Availability

All data were deposited at the link in the Appendix A.

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
