# Peer review of "Predicting the 14-Day Hospital Readmission of Patients with Pneumonia Using Artificial Neural Networks (ANN)"

_ijerph, 2021, doi:10.3390/ijerph18105110_

Round 1
Reviewer 1 Report
The article presents a prediction model of UPRA using ANN. The article is well written and structured. The authors show the rationale of the research and the contribution made to the state of art.
The material and methods employed are broadly explained. However, it is not clear how the train/test splits have been done in the four scenarios. It seems that the test set is extracted from the data set used for training, i.e. first scenario: training set n=21,892 (the entire data set), test set n=6,568 (0.3 of the training set). If that is the case, there is a misconception in the use of train/test set because it seems that samples used for testing have been previously employed for training. In the test phase, the model must be performed over new unseen data. The authors must make clear this issue since it affects the classification performance and the results section. This point is the reason why a major revision is requested.
In addition, the results are thoroughly described especially in section 3.1. However, classification results shown in section 3.2 are confusing because the models implemented in the research (ANN and CNN) presents a worse performance in accuracy in comparison to the models provided by the Weka tool. Thus, the decision of taking ANN and CNN should be justified.
The discussion and discussion are well addressed and the approach adopted is quite comprehensive.
Other details to be corrected:
line 177: acronym of NIQJ must be defined
Author Response
Reviewer1:
The article presents a prediction model of UPRA using ANN. The article is well written and structured. The authors show the rationale of the research and the contribution made to the state of art.
The material and methods employed are broadly explained. However, it is not clear how the train/test splits have been done in the four scenarios. It seems that the test set is extracted from the data set used for training, i.e. first scenario: training set n=21,892 (the entire data set), test set n=6,568 (0.3 of the training set). If that is the case, there is a misconception in the use of train/test set because it seems that samples used for testing have been previously employed for training. In the test phase, the model must be performed over new unseen data. The authors must make clear this issue since it affects the classification performance and the results section. This point is the reason why a major revision is requested.
Response: Advised by the reviewer, we remade Table 1 in comparison of the two models merely on the results of using the training sample to predict the testing sample in the revised manuscript. .
In addition, the results are thoroughly described especially in section 3.1. However, classification results shown in section 3.2 are confusing because the models implemented in the research (ANN and CNN) presents a worse performance in accuracy in comparison to the models provided by the Weka tool.
Response: In the revised version, we see the results in WeKa displaying aberrant results due to using the imbalanced data based on the maximum accuracy. As such, resulting in imbalanced SENS and SPEC was observed.
In contrast, we applied a scheme in estimation of parameters using the average(residuals in URPA) + average(residuals in Non-URPA) to overcome the phenomenon of imbalanced SENS and SPEC.
Thus, the decision of taking ANN and CNN should be justified.
Response: Based on the previous question, we justify the reason for using the ANN in this study.
It is worth noting that the accuracies in WeKA fail to construct a balanced SENS and SPEC. The high accuracy is problematic and unreliable due to a tendency toward the Non-UPRA classification for all cases (i.e., none was classified into UPRA due to the imbalanced-class numbers existed in data. As such, the high accuracy in Weka is meaningless. We should consider the composite score of AUC in evaluation of model validation.); see Figure 7.
The discussion and discussion are well addressed and the approach adopted is quite comprehensive.
Response: We have made improvement according to the second reviewer’s suggestion to make the revised manuscript more readable and comprehensive than the original version of manuscript.
Other details to be corrected:
line 177: acronym of NIQJ must be defined
Response: The typo of NIQJ has been replaced with UPRA in the whole context.

Reviewer 2 Report
In this article, the authors aimed to build a prediction model for the early detection of 14-dayUPRA) of patients with pneumonia.
The issue chosen by the authors is of particular importance to society as a whole. Their methodology is valid and, thus, reliable results obtained.
The biggest problem in this article lies in the presentation. The use of English by the authors is at a high level.
The article needs several improvements and corrections to be made by the authors.
1)First of all, I would like to point out that the chosen font, by the authors, is very tedious to the reader and, also, it is different from the abstract.
2)In my opinion, the paper title is too long. Choose a short and appropriate title without details.
3)Abstract has many unnecessary details, e.g. Weka. Also, it has so many highlights in numbers that tire the reader. Specifically, the abstract section should be rewritten differently, following the structure.1)background, 2)motivation, 3) gap challenges, 4) proposed approach, 5) evaluation and results from 6) significance.
4)The introduction section needs to be redefined. There's no need for so many sub-sections. Although the article contains many references, they are not appropriately supported.
Try to write a single section and emphasize, briefly, the main contribution of the work.
5)I suggest you add a separate section entitled "Related Work" after the introduction section to transit smoothly in the following parts of the work. Make a proper discussion and comparison with the state-of-the-art.
6)Organize the section Materials and Methods better. Give more technical details of your methods. Is it necessary to have so many subsections here?
7)Kindly reduce the resolution of figure 2 to half and align it in the centre.
8)In section 4.2, it does not make sense with so many numbers
9)The authors miss the experiment setup. Please, demonstrate the environment of the experiments in detail.
10)The way you present the results is not attractive. You should summarize the results (i.e., Sensitivity, Specificity, Accuracy) in a table. Which techniques do you prefer over the previous ones?
11)The conclusion section is roughly written. Here, you failed to highlight the results in the best way. Αlso, there is no numbering in this section.
In general, the text organization should be written from the beginning. Authors should emphasize the presentation of their work.
The technical contribution is limited. The authors have applied already known techniques and methods.
Author Response
Reviewer 2:
In this article, the authors aimed to build a prediction model for the early detection of 14-dayUPRA) of patients with pneumonia.
The issue chosen by the authors is of particular importance to society as a whole. Their methodology is valid and, thus, reliable results obtained.
The biggest problem in this article lies in the presentation. The use of English by the authors is at a high level.
The article needs several improvements and corrections to be made by the authors.
- First of all, I would like to point out that the chosen font, by the authors, is very tedious to the reader and, also, it is different from the abstract.
Response: We have rewritten Abstract and made the fonts used in Abstract in adherence with the journal’s guideline accordingly. Notably, the font in Abstract is surely different from the context based on the template provided by the journal.
- In my opinion, the paper title is too long. Choose a short and appropriate title without details.
Response: Advised by the reviewer, we have shortened the title as “Predicting the 14-Day Hospital Readmission of Patients with Pneumonia Using Artificial Neural Networks (ANN)”
- Abstract has many unnecessary details, e.g. Weka. Also, it has so many highlights in numbers that tire the reader. Specifically, the abstract section should be rewritten differently, following the structure.1)background, 2)motivation, 3) gap challenges, 4) proposed approach, 5) evaluation and results from 6) significance.
Response: In terms of the limitation of word number in Abstract limited within 200, we have rewritten the Abstract and removed those non-sense numbers in Abstract based on the suggested structure provided by the reviewer.
- The introduction section needs to be redefined. There's no need for so many sub-sections. Although the article contains many references, they are not appropriately supported.
Try to write a single section and emphasize, briefly, the main contribution of the work.
Response: (1) As suggested by the reviewer, we have rewritten the section of Introduction to make the whole concept and research demand clearly described in this section.
(2) the main contributions of the work were addressed in the section of Contributions from this Study in Discussion.
5)I suggest you add a separate section entitled "Related Work" after the introduction section to transit smoothly in the following parts of the work. Make a proper discussion and comparison with the state-of-the-art.
Response: As advised by the reviewer, we added a subtitle of 1.1. Related Work to smoothly transit the following four subsections.
In Discussion, we add one paragraph to emphasize the study on the advantage and disadvantage when compared to the other counterparts in academics.
- Organize the section Materials and Methods better. Give more technical details of your methods. Is it necessary to have so many subsections here?
Response: As suggested by the reviewer, we have reorganized the section of Methods and simplified them to an acceptable necessity but not too-much to tire readers. Only five subsections are in Methods, including (1) data sources, (2) variables extraction, (3) model building, (4) Tasks to work in results, and (5) Statistical tools.
- Kindly reduce the resolution of figure 2 to half and align it in the centre.
Response: We have reduced the resolution of Figure 2 and align it in the center according to the reviewer’s suggestion.
- In section 4.2, it does not make sense with so many numbers
Response: We have removed them in the revised version and classified them into the literature reviews in feature variables in the revised version of manuscript.
- The authors miss the experiment setup. Please, demonstrate the environment of the experiments in detail.
Response: We have added an MP4 video to demonstrate the whole steps in detail in Appendix B at https://www.youtube.com/watch?v=Xj9pJMxfs0o.
- The way you present the results is not attractive. You should summarize the results (i.e., Sensitivity, Specificity, Accuracy) in a table. Which techniques do you prefer over the previous ones?
Response: (1) We have made the improvement in data presentation in the revised version according to the reviewer’s suggestion. Table 1 has been reproduced in the revised version based on the three indicators of SENS, SPEC, and Accuracy, besides the AUC frequently used in comparison of prediction models.
(2) the technique regarding the overrated accuracy mentioned in imbalanced-class data was particularly illustrated in this study.
- The conclusion section is roughly written. Here, you failed to highlight the results in the best way. Αlso, there is no numbering in this section.
Response: We have rewritten the section of Conclusion to highlight the results briefly. Moreover, the numbering in conclusion has been removed from the revised version.
In general, the text organization should be written from the beginning. Authors should emphasize the presentation of their work.
The technical contribution is limited. The authors have applied already known techniques and methods.
Response: We have rewritten the text necessary for avoiding jargons and technical terms in the revised version. The data presentations have been also considered in the revised version to make the data clearly understood than the previous version of manuscript.

Round 2
Reviewer 1 Report
The authors have addressed the main comment made for the major revision about train/test split methods. Moreover, they have improved some sections of the article specifying clearer certain aspects of methods follows. Then, the paper should be accepted in its current form.
